# Evaluating Antitumor and Antioxidant Activities of Yellow *Monascus* Pigments from *Monascus ruber* Fermentation

**DOI:** 10.3390/molecules23123242

**Published:** 2018-12-07

**Authors:** Hailing Tan, Ziyi Xing, Gong Chen, Xiaofei Tian, Zhenqiang Wu

**Affiliations:** 1School of Biology and Biological Engineering, South China University of Technology, Guangzhou 510006, China; callingmeling@163.com; 2Pan Asia (Jiangmen) Institute of Biological Engineering and Health, Jiangmen 529080, China; 3The Key Laboratory of Pathobiology, Ministry of Education, The College of Basic Medical Sciences, Jilin University, Changchun 130021, China; xingzy17@mails.jlu.edu.cn; 4School of Environmental Ecology and Biological Engineering, Wuhan Institute of Technology, Wuhan 430205, China; chengong@wit.edu.cn

**Keywords:** natural yellow *Monascus* pigments, water-soluble, antioxidation, migration, invasion, MCF-7 cells

## Abstract

Yellow *Monascus* pigments can be of two kinds: Natural and reduced, in which natural yellow *Monascus* pigments (NYMPs) attract widespread attention for their bioactivities. In this study, the antioxidative and antibreast cancer effects of the water-soluble NYMPs fermented by *Monascus ruber* CGMCC 10910 were evaluated. Results showed that water-soluble NYMPs had a significantly improved antioxidative activities compared to the reduced yellow *Monascus* pigments (RYMPs) that were chemically derived from orange or red *Monascus* pigments. Furthermore, NYMPs exhibited a concentration-dependent inhibition activity on MCF-7 cell growth (*p* < 0.001). After a 48-h incubation, a 26.52% inhibition yield was determined with 32 μg/mL of NYMPs. NYMPs also significantly inhibited the migration and invasion of MCF-7 cells. Mechanisms of the activities were associated with a down-regulation of the expression of matrix metalloproteinases and vascular endothelial growth factor. Rather than being alternatively used as natural colorants or antioxidants, this work suggested that NYMPs could be selected as potential functional additives in further test of breast cancer prevention and adjuvant therapy.

## 1. Introduction

*Monascus* metabolites are popular bioactive candidates for the development of functional foods. *Monascus*-fermented pigments have shown bioactivities of antioxidation [1,2] and anticancer [3]. Natural yellow *Monascus* pigments (NYMPs) have attracted great attention due to their beneficial effects for human healthcare [1,2,4]. Alcohol-soluble NYMPs, such as ankaflavin [5], monascin [6], monascuspiloin [7], monapurpyridine A [8], and monaphilones A [9], are reported with remarkable anticancer activities. Ho et al. [10] found that ankaflavin, synergized with Monacolin K, inhibited proliferation and induced apoptosis in mouse lung cancer LCC cells. Additionally, Shi et al. [11] reported that the monascin increased the survival of *Caenorhabditis elegans* under juglone-induced oxidative stress and attenuated endogenous levels of reactive oxygen species. Although NYMPs possess promising biological activities, the hydrophobic property of most NYMPs limits their further application in the food industry. As an important food coloration additive, water-soluble NYMPs are chemically derived from the hydrophobic orange *Monascus* pigments. It was reported that some water-soluble NYMPs could be produced through fermentation with *Monascus ruber* CGMCC 10910 [12], which have a great potential to be used as additives in functional food and healthcare products. However, the knowledge of the biological activities of water-soluble NYMPs needs to be further expanded.

Breast cancer is one of the leading causes of mortality among women worldwide [13]. The current clinical treatments of breast cancer usually come with serious adverse effects [14]. Therefore, it is necessary to develop effective antitumor agents with reduced side effects to treat breast cancer. Drugs from natural resources have shown possible anticancer effects, with mechanisms of inhibiting angiogenesis [15], decreasing cell growth and proliferation, apoptosis [16], and preventing oxidation [17].

Several lines of evidence have established that antioxidation attenuates free radical-induced oncogene mutation and prevents carcinogenesis [18]. What is more, the inhibition ability of matrix metalloproteinases (MMPs) is important for the prevention of cell invasion [19]. For breast cancer cells, the vascular endothelial growth factor (VEGF) acts as the survival factor which possesses the ability to promote cancer growth [20,21]. Reducing the expressions of MMPs and VEGF is one of the promising approaches in cancer therapy.

In this work, the antioxidative effects of water-soluble NYMPs from *M. ruber* CGMCC 10910 were studied. The antitumor activity of the NYMPs was determined by exploring their effects on cell migration and invasion of the MCF-7 cell line. In the meantime, the possible underlying mechanisms in vitro models were also investigated.

## 2. Results and Discussion

### 2.1. Antioxidant Activities of Water-Soluble NYMPs

Previous work has verified that water-soluble NYMPs could be produced by the *M. ruber* CGMCC 10910 without citrinin [12]. The main extracellular pigments were mixtures of four components, including Y1, Y2, Y3, and Y4, with molecular weights of 250, 254, 402, and 358 Da., respectively. Huang et al. [22] reported that the chemical characteristics of Y1, Y2, Y3, and Y4, in which Y3 and Y4 had strong yellow fluorescence at UV of 365 nm and Y1 was the Azanigerone E (C_13_H_14_O_5_), a known natural pigment.

The antioxidant activities of water-soluble NYMPs were assessed by DPPH and ABTS^+^ (2,2′-azino-bis(3-ethylbenzothiazoline-6-sulfonic acid)) scavenging assays. Butylated hydroxytoluene (BHT) was used as a positive control. DPPH scavenging occurred by the reduction of DPPH radical in the presence of a proton-donating substance. The DPPH radical scavenging activities were promoted along with increased NYMP concentrations (Table 1). When the concentration of 1 mg/mL was tested, the DPPH scavenging ratio by the NYMPs was 86.33%, which was close to that of the BHT but 2.63 times that of the reduced yellow *Monascus* pigments (RYMPs) with a ratio of only 32.84%. Compared to the RYMPs, the NYMPs had a more effective DPPH scavenging activity, since the calculated IC_50_ of NYMPs and RYMPs were 0.23 mg/mL and 2.77 mg/mL, respectively.

The ABTS^+^ scavenging assay was generally used for evaluating the antioxidant activities of hydrophilic compounds. The ABTS^+^ radical scavenging ratio by 0.25 mg/mL NYMPs was equivalent to that of the BHT (Table 2). The IC_50_ of the NYMPs was 0.035 mg/mL, which was far lower than 0.176 mg/mL of the RYMP. This indicated that the *Monascus*-fermented NYMPs possessed significantly improved antioxidative activities than the chemically-derived RYMPs.

Srianta et al. [23] reported that a higher *Monascus* pigment content had higher antioxidant activities. Similarly, the antioxidant activities of NYMPs applied corresponded to the intensity of the pigments in this work. To our knowledge, the antioxidant activities of *Monascus*-fermented substrate were generally contributed by the water-soluble *Monascus* pigments [24,25]. The advantage of water-soluble NYMP products showed a considerable application prospect in the food industry.

### 2.2. Cytotoxicity of Water-Soluble NYMPs on MCF-7 Cells

Water-soluble yellow *Monascus* pigments have attracted much attention thanks to their low toxicity as well as improved antioxidative activity, which could potentially be used to prevent carcinogenesis in human body. Ankaflavin was found to be toxic to human cancer cell lines Hep G2 and A549, while it displayed no significant toxicity to normal MRC-5 and WI-38 cells [26].

The toxicity of fermented NYMPs (0–128 μg/mL) on cell viability was determined by the microculture tetrazolium (MTT) assay. It was found that NYMPs exhibited no significant toxicity to MCF-7 cells. As shown in Figure 1, cell viability was not significantly affected by NYMPs after a 24-h treatment, which indicated that the NYMPs were not toxic to MCF-7 cells. The result was consistent with the yellow *Monascus* pigment, Monascusone A, which exhibited no cytotoxicity against breast cancer [27].

Several studies showed that *Monascus* metabolites had antiproliferative activities in various cancers [28]. Hsu et al. [29] investigated four new azaphilones with yellow fluorescence, which exhibited antiproliferative activities against human laryngeal carcinoma (HEp-2) and human colon adenocarcinoma (WiDr). Chang et al. [27] reported that the extract of *Monascus purpureus* CWT715 significantly inhibited the proliferation of SK-Hep-1 cells at 100 μg/mL. Park et al. [30] found that RYRG extracts had antiproliferative effects against HepG2 human liver cancer cells, HT-29 human colon cancer cells, and B16F10 murine melanoma cells. Figure 2 shows that the proliferation of MCF-7 cells is inhibited by NYMPs in a concentration-dependent way by 23.56% (*p* < 0.001) and 26.52% (*p* < 0.001) at 32 μg/mL in a 24-h and 48-h incubation, respectively. The results suggested the existence of antiproliferative activity of the water-soluble NYMPs. As Huang et al. [22] reported, the Y3 and Y4 components had strong yellow fluorescence in the NYMPs. We speculated that Y3 and Y4 might be the active ingredients in the NYMPs.

Figure 3 shows that NYMPs have significantly inhibited the cell invasion of MCF-7 cells at a 24-h incubation, with 31.01% (*p* < 0.01) inhibition yield at a concentration of 40 μg/mL. Similar to those on cell invasion in the Transwell assays, NYMPs had a 32.58% (*p* < 0.01) inhibition on cell migration at a concentration of 40 μg/mL (Figure 4). In the wound healing assay, the reduction of the transferred area indicated that the migration inhibition yield of MCF-7 cells by NYMPs (40 μg/mL) was 40.44% (*p* < 0.01) (Figure 5). The results suggested that NYMPs inhibited the migration and invasion of MCF-7 cells.

### 2.3. Regulating Effect of Water-Soluble NYMPs on the Expression of MMP-2, MMP-9, and VEGF

MMPs were required for extravasation out of the vessel, leading to the movement of cancer cells to the target tissue, which could work as pivotal targets for suppressing breast cancer invasion and metastasis [31]. Known as key enzymes in the degradation of type IV collagen, MMP-2 [32] and MMP-9 [33] were overexpressed in breast cancer cells among all of the MMPs [34]. The inhibition of MMP-2 and MMP-9 expressions was a critical step in the prevention of cancer metastasis. To examine the possible antimetastatic mechanisms of NYMPs, the expression of MMP-2 and MMP-9 in the culture media of MCF-7 cells was determined by Western blotting assay. The NYMPs significantly suppressed the expression of MMP-2 by 23.42% (*p* < 0.01), and slightly decreased the activities of MMP-9 by 7.53% at a concentration of 40 μg/mL (Figure 6). The results indicated that the possible antimetastatic mechanism of NYMPs was through the inhibition of the expression of MMP-2 protein.

Monascin and ankaflavin targeted the network of pathogenic inflammatory mediators, which contributed in tumor angiogenesis and metastasis. A significant reduction in the VEGF level was reported as a marker of angiogenesis in the free yellow-*Monascus*-pigment-treated mice group compared to the positive control group [35]. Water-soluble NYMPs significantly suppressed the expression of VEGF by 39.72% (*p* < 0.001) at a concentration of 40 μg/mL (Figure 6). This implied that NYMPs at least in part contributed to an anticancer effect directly by regulating VEGF protein expression.

## 3. Materials and Methods 

### 3.1. Microorganisms and Chemicals

The *M. ruber* 10910 strain preserved in the China General Microbiological Culture Collection Center (CGMCC, Beijing, China) was cultivated on potato dextrose agar (PDA) medium at 30 °C for 7 days before use.

DPPH was purchased from Shanghai Macklin Biochemical Co., Ltd. (Shanghai, China). ABTS^+^ and BHT were obtained from Aladdin Industrial Corporation (Shanghai, China). All chemical reagents were purchased from ZhiYuan Reagent Co., Ltd. (Tianjin, China) in analytical grade. RYMPs samples prepared by chemical derivation from red *Monascus* pigment were purchased from Tianyi Biotechnology Co., Ltd. (Dongguan, Guangdong, China).

### 3.2. Monascus Pigment Fermentation

The method for *Monascus* pigment fermentation was according to the approach reported by Wang et al. [12]. A total of 5–6 loops of single colonies (10 mm diameter) were scraped off from the culture plate and inoculated in a 250 mL flask with 50 mL seed medium and incubated at 30 °C for 25 h in a rotary shaker (Labwit Scientific, Shanghai, China) at 180 rpm. For fermentation, 2 mL seed culture was transferred to a 250 mL Erlenmeyer flask containing 25 mL fermentation culture medium and incubated at 30 °C for 9 days on a shaker in 180 rpm. The preparation of seed and fermentation culture media is referred to in our previous work [22].

### 3.3. Isolation of Water-Soluble NYMPs

The fermented broth was filtered through a mixed cellulose esters membrane (0.8 mm, Xiya purification equipment Co. Ltd., Shanghai, China). According to Wu et al. [36], with minor modifications, a column (30 × 6 cm, 1.98 L) with 150 g adsorption resin (DA-201C, Zhengzhou Qinshi Technology Co., Ltd., Henan, China), was employed for isolation and purification of the NYMPs from the filtrated broth. A 60% (*v*/*v*) ethanol solution was used as the eluent with a flow rate of 1 mL/min. After eluting for 90 min, the aqueous ethanol solution was submitted to vacuum-evaporation and freeze-dried. The solid was dispersed in distilled water and centrifuged at 8000× *g* for 20 min. The supernatant was collected as the NYMPs solution.

### 3.4. Antioxidant Activity Determination

The antioxidant activities of the yellow *Monascus* pigments were determined by the activities of DPPH radical scavenging and ABTS radical cation (ABTS^+^) scavenging evaluated. According to the method reported by Wang et al. [37], with minor modifications, 50 μL NYMPs solution at concentrations of 25, 75, 100, 500, and 1000 μg/mL were mixed with 200 μL DPPH methanol solution (600 μmol/L). The mixture was kept in the dark at room temperature for 60 min. Absorbance at 517 nm (A_517_) was measured using a SpectraMax Gemini microtiter plate reader (Molecular Devices, Sunnyvale, CA, USA) with Softmax Pro 3.0 software (Molecular Devices, Sunnyvale, CA, USA). BHT was used as a positive control. The sample concentration achieving a 50% DPPH free radical scavenging activity was defined as the IC_50_ value. The efficiency for DPPH free radical scavenging was calculated according to Equation (1): (1)$$\mathrm{DPPH}\;\mathrm{radical}\;\mathrm{scavenging}\;\mathrm{activity}\;\left(\%\right)=1-\frac{A_{S}}{A_{0}}\times 100$$
where A_0_ is the A_517_ of the DPPH solution and As is the A_517_ of the mixture after reaction.

The freshly prepared ABTS^+^ solution was prepared by oxidation of 7 mmol/L ABTS with 2.45 mmol/L potassium persulfate [37]. A 50 μL sample solution at concentrations of 25, 50, 75, 100, and 250 μg/mL was thoroughly mixed with 200 μL of ABTS^+^ solution and kept at room temperature for 60 min in the dark. The absorbance at 734 nm (A_734_) was measured. BHT was used as a positive control. The ABTS^+^ scavenging activity was calculated by Equation (2):(2)$${\mathrm{ABTS}}^{+}\mathrm{radical}\;\mathrm{scavenging}\;\mathrm{activity}\;\left(\%\right)=\left(1-\frac{A_{2}}{A_{1}}\right)\times 100$$
where A_1_ is the A_734_ of the ABTS^+^ solution and A_2_ is the A_734_ of the mixture after reaction.

### 3.5. MCF-7 Cell Line and Culture

Human breast cancer cells MCF-7 were obtained from the key laboratory of pathobiology, Ministry of Education, Jilin University (Changchun, Jilin, China), and routinely cultured in H–DMEM (High Glucose- Dulbecco’s Modified Eagle Medium) containing 10% (*v*/*v*) fetal bovine serum (FBS), 0.37% (*w*/*v*) NaHCO_3_, penicillin (100 unit/mL), and streptomycin (100 unit/mL) in a humidified incubator (Thermo Fisher Scientific, Waltham, MA, USA) under 5% CO_2_ and 95% air at 37 °C.

### 3.6. Measurement of Cytotoxicity and Inhibition Activity in Cell Proliferation

The cytotoxic effects of the various concentrations of NYMPs on MCF-7 cells were determined by MTT assay [19]. Cells (4 × 10^3^ per well) were seeded in triplicate onto a sterile 96-well plate (Corning, Corning, NY, USA) containing 100 μL medium in each well. After a 24-h incubation, the medium was replaced with 100 μL fresh FBS-free medium which contained NYMPs at concentrations of 0, 2, 4, 8, 16, 32, 64, and 128 μg/mL. The plate was incubated for 24 h and then the media were discarded. Afterwards, the cells were stained with 100 μL of MTT solution at 37 °C for 4 h. Thereafter, the supernatant was aspirated, and 150 μL of DMSO was added to dissolve the formazan. The optical density at 570 nm (OD_570_) was determined.

The inhibition activities of NYMPs toward MCF-7 cells proliferation were quantified using the MTT assay as described above, with minor modifications. Briefly, 4 × 10^3^ MCF-7 cells were plated in each well of the 96-well plate and incubated at 37 °C in 5% CO_2_ for 24 h. Then, the cells were treated with various doses of NYMPs at 37 °C in 5% CO_2_ for 24 h and 48 h. (Medium: H-DMEM with 10% FBS). The remaining cells on the bottom of wells were quantified. The percentage of inhibition was calculated using Equation (3):(3)$$\mathrm{Growth}\;\mathrm{inhibition}\;\left(\%\right)=\left(1-\frac{{\mathrm{OD}}_{\mathrm{test}}}{{\mathrm{OD}}_{\mathrm{control}}}\right)\times 100$$
where OD_control_ is the OD_570_ in the control wells (cells incubated with vehicle only) and OD_test_ is the OD_570_ of the cells exposed to NYMPs.

### 3.7. Transwell Assay

Cell migration was assayed according to the methods of Justus et al. [38], with some modifications. MCF-7 cells were suspended in 200 μL DMEM (serum free supplemented with NYMPs), placed in the upper transwell chambers, and 600 μL medium with 20% FBS was added into the lower chamber. Before the determination of cell migration and cell invasion, the number of cells was adjusted to an equal number (2 × 10^5^ cells/mL) in the Transwell assays to avoid interference by cell numbers. After 24 h of incubation at 37 °C, the cells on the upper surface of the filter were completely wiped away with a cotton swab. The cells on the lower surface of the filter were fixed in 4% paraformaldehyde, stained with crystal violet, and then counted under a microscope (Olympus Corporation, Tokyo, Japan). For each replicate, the tumor cells in 10 randomly-selected fields were determined, and the counts were averaged.

### 3.8. Wound-Healing Assay

Cells were seeded in 2 × 10^5^ cells/mL and grown to 80–90% confluence in a 6-well plate at 37 °C. The monolayers were scratched with a 200 μL sterile pipette tip, washed twice with PBS, and then replaced with complete DMEM. MCF-7 cells were treated with NYMPs (0, 10, and 40 μg/mL) and incubated for 24 h. Cell migration into the wound area was photographed under an inverted microscope (Olympus Corporation, Tokyo, Japan). Migrated cells across the blue lines were calculated in 3 random fields from each triplicate treatment.

### 3.9. Invasion Assay

Cell invasion was analyzed using Transwell assays [38], with some modifications. The upper chambers were coated with 60 μL Matrigel (dilution 1:6) in cold DMEM and 4 × 10^4^ cells were then seeded in the upper chamber of each well in serum-free medium containing NYMPs. Additionally, 600 μL DMEM media supplemented with 20% FBS was added to the lower chamber. The cells were incubated for 24 h. The conditions and methods of staining and counting were similar with the migration assay.

### 3.10. Western Blotting

Total cellular proteins extracted from cells were assayed using a BCA (Bicinchoninic acid) kit (Beyotime Institute of Biotechnology, Beijing, China). Equal amounts of protein were subjected to sodium dodecyl sulfate polyacrylamide gel electrophoresis and then electrophoretically transferred to polyvinylidene fluoride membranes. Membranes were blocked with 5% nonfat dry milk in tris buffered saline (TBS) with 0.1% Tween-20 for 1 h and incubated with anti-MMP 2 (abcam, Cambridge, UK, ab92536, 1:1000), anti-MMP 9 (abcam, ab76003, 1:1000), anti-VEGFA (Wanleibio, Shenyang, China, WL0009b, 1:1000,) and anti-GAPDH (Glyceraldehyde-3-phosphate dehydrogenase) (CST, Danvers, MA, USA, #2118, 1:1000) antibodies at 4 °C overnight. The protein blots were then visualized with a use of the streptavidin–HRP ELC (HRP: Horseradish Peroxidase; ELC: Electrochemiluminescence) assay (Beyotime Institute of Biotechnology, Beijing, China). Image-Pro Plus 6.0 (Media Cybernetics, Rockville, MD, USA) was used for the quantification of the data from western blot.

### 3.11. Statistical Analysis

All data were expressed as the mean ± standard deviations (SD) from at least triplicate independent runs. Statistical analysis was performed with SPSS Statistics 17 (SPSS Inc., Chicago, IL, USA). Statistical differences were analyzed using t-tests and one-way analysis of variance (ANOVA). A value of *p* < 0.05 was considered as significant difference.

## 4. Conclusions

Water-soluble NYMPs exhibited the capacity of antioxidation and inhibited the migration and invasion of breast cancer MCF-7 cells with no cell cytotoxicity. The antitumor effect is possibly associated with the down-regulation of MMP-2 and VEGF protein expression. Cell line studies suggest that the fermented water-soluble NYMPs could be selected as potential functional additives in further tests of breast cancer prevention and adjuvant therapy.

## Figures and Tables

**Figure 1 molecules-23-03242-f001:**
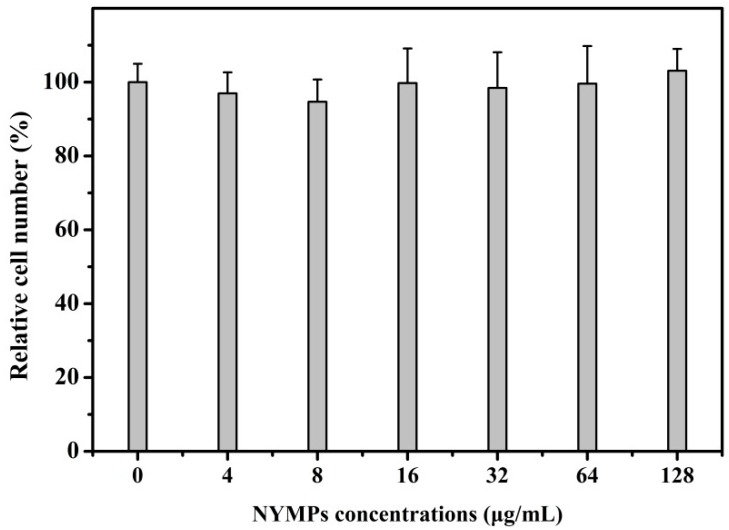
Cytotoxicity of water-soluble NYMPs on human breast cancer MCF-7 cells under different concentrations. Each value represents the mean ± standard deviation (*n* = 3).

**Figure 2 molecules-23-03242-f002:**
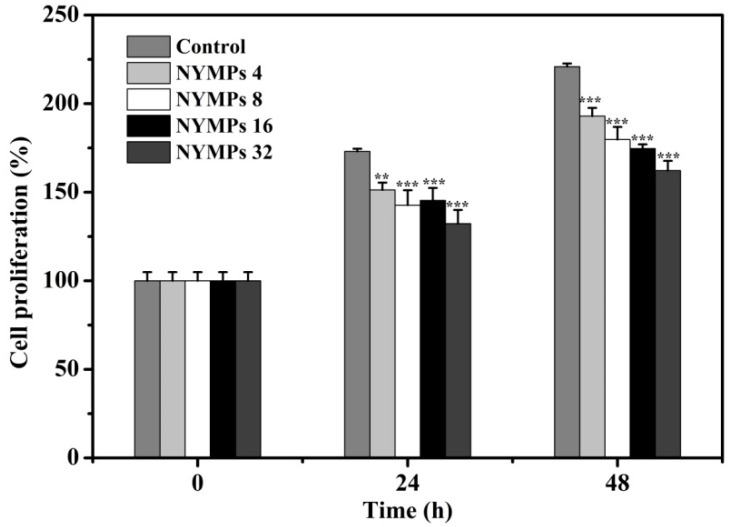
Effects of water-soluble NYMPs on cell proliferation of human breast cancer MCF-7 cells. Cells were incubated with NYMPs for 24 and 48 h. NYMPs 4, NYMPs 8, NYMPs 16, and NYMPs 32 represents 4, 8, 16, and 32 μg/mL, respectively. Each value represents the mean ± standard deviation (*n* = 3). * *p* < 0.05, ** *p* < 0.01 and *** *p* < 0.001 indicated statistically significant differences versus control group.

**Figure 3 molecules-23-03242-f003:**
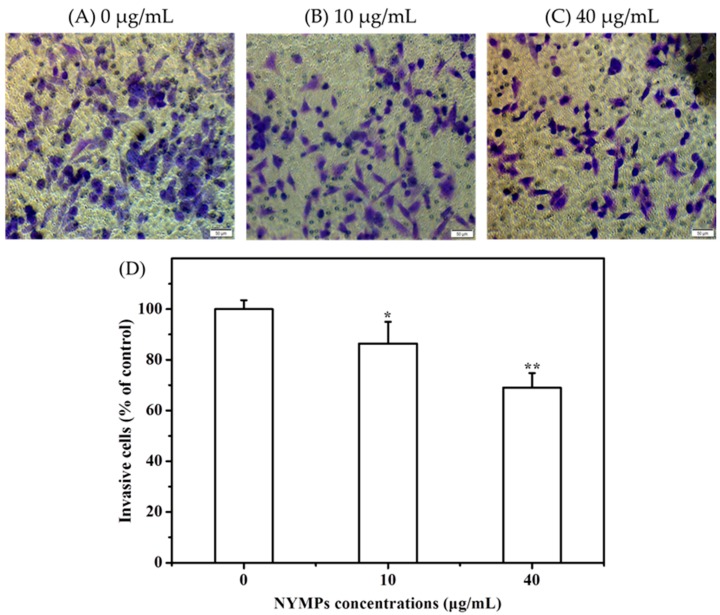
Effects of NYMPs on cell invasion of human breast cancer MCF-7 cells (Magnification× 200). (**A**) Images of cells treated without NYMPs. (**B**) Image of cells treated with 10 μg/mL NYMPs. (**C**) Image of cells treated with 40 μg/mL NYMPs. (**D**) Invasion ability. Each value represents the mean ± standard deviation (*n* = 3). * *p* < 0.05 and ** *p* < 0.01 indicated statistically significant differences versus control group.

**Figure 4 molecules-23-03242-f004:**
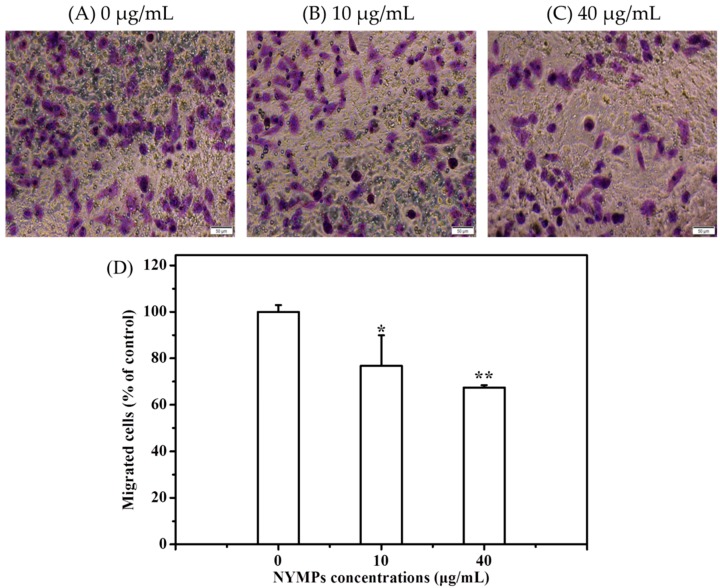
Effects of NYMPs on cell migration of human breast cancer MCF-7 cells (Magnification× 200). (**A**) Image of cells treated without NYMPs. (**B**) Image of cells treated with 10 μg/mL NYMPs. (**C**) Image of cells treated with 40 μg/mL NYMPs. (**D**) Invasion ability. Each value represents the mean ± standard deviation (*n* = 3). * *p* < 0.05 and ** *p* < 0.01 indicated statistically significant differences versus control group.

**Figure 5 molecules-23-03242-f005:**
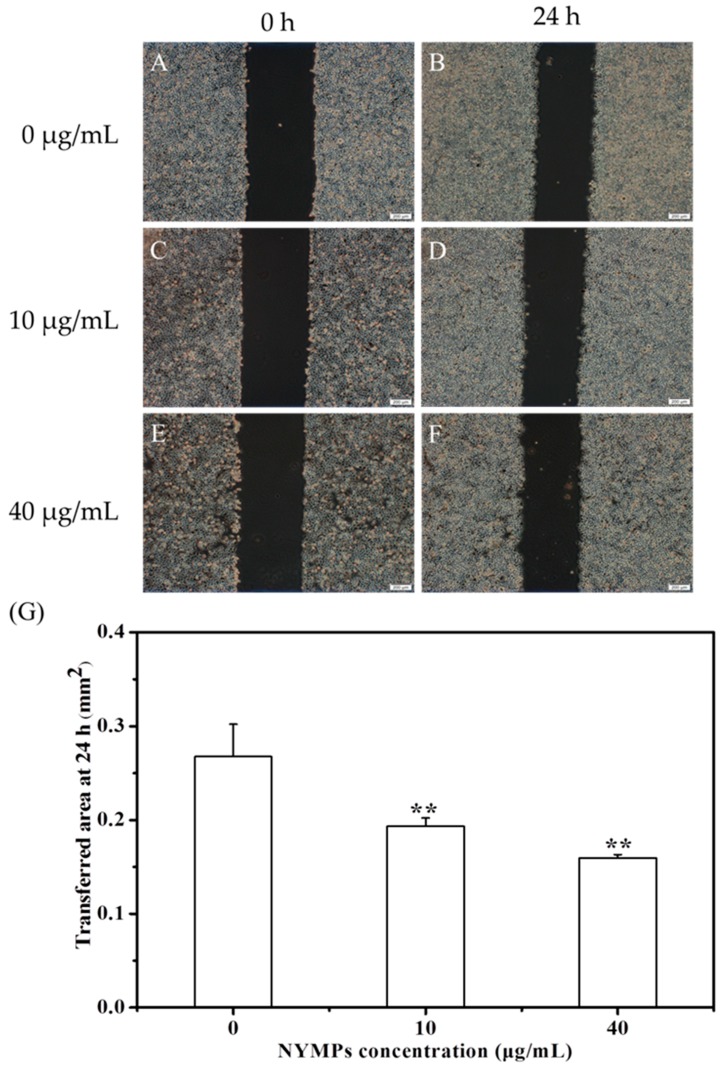
Effects of NYMPs on cell migration of human breast cancer MCF-7 cells (Magnification× 40). (**A**,**B**) Images of cells treated without NYMPs. (**C**,**D)** Images of cells treated with 10 μg/mL NYMPs. (**E**,**F**) Images of cells treated with 40 μg/mL of NYMPs. (**G**) Migrated ability. Each value represents the mean ± standard deviation (*n* = 3). ** *p* < 0.01 indicated statistically significant differences versus control group.

**Figure 6 molecules-23-03242-f006:**
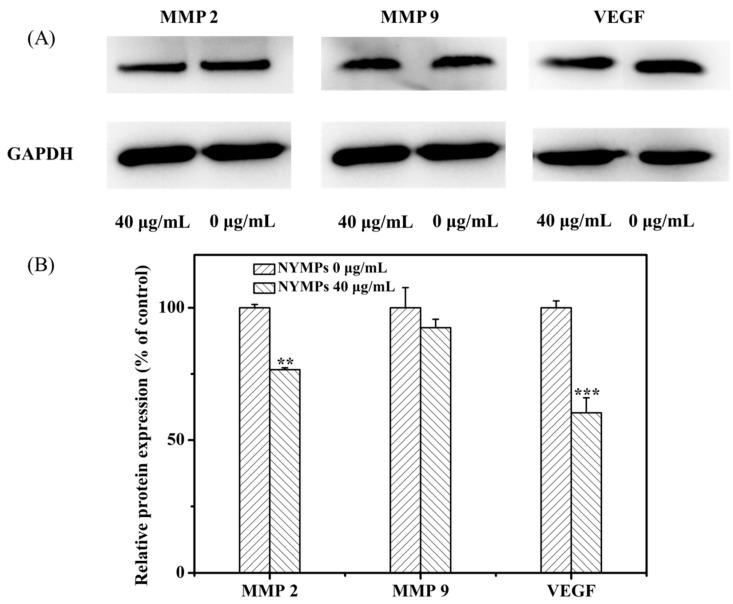
Effects of NYMPs on protein expression level of MMP-2, MMP-9, and VEGF. (**A**) Expression of MMP-2, MMP-9, and VEGF, GAPDH (Glyceraldehyde-3-phosphate dehydrogenase) as loading control; (**B**) Densitometric analyses of (A). Each value represents the mean ± standard deviation (*n* = 3). ** *p* < 0.01, *** *p* < 0.001 indicated statistically significant differences versus control group.

**Table 1 molecules-23-03242-t001:** DPPH radical scavenging activity of yellow *Monascus* pigments.

Samples (mg/mL)	0.025	0.075	0.100	0.500	1.000
RYMPs ^b^	^d^ 3.59 ± 0.14	6.57 ± 0.75	8.91 ± 0.19	20.75 ± 0.13	32.84 ± 1.02
NYMPs ^a^	9.54 ± 0.23	21.23 ± 1.37	26.74 ± 0.36	69.03 ± 0.92	86.33 ± 1.50
BHT ^c^	14.25 ± 1.45	37.48 ± 2.63	47.07 ± 2.40	94.42 ± 0.17	95.14 ± 0.58

^a^ NYMPs represents natural yellow *Monascus* pigments. ^b^ RYMPs represents reduced yellow *Monascus* pigments. ^c^ BHT represents butylated hydroxytoluene. ^d^ Each value represents averages of three replicates ± standard deviation.

**Table 2 molecules-23-03242-t002:** ^e^ ABTS^+^ radical scavenging activity of yellow *Monascus* pigments.

Samples (mg/mL)	0.025	0.050	0.075	0.100	0.250
RYMPs ^b^	^d^ 13.95 ± 0.94	25.33 ± 0.32	38.00 ± 0.86	45.77 ± 0.33	84.92 ± 1.77
NYMPs ^a^	38.00 ± 1.71	60.67 ± 1.00	81.92 ± 0.33	87.03 ± 0.11	99.95 ± 0.09
BHT ^c^	94.42 ± 2.48	99.70 ± 0.38	99.45 ± 0.31	99.76 ± 0.28	98.61 ± 0.10

^a^ NYMPs represents natural yellow *Monascus* pigments. ^b^ RYMPs represents reduced yellow *Monascus* pigments. ^c^ BHT represents butylated hydroxytoluene. ^d^ Each value represents averages of three replicates ± standard deviation. ^e^ ABTS represents 2,2′-azino-bis(3-ethylbenzothiazoline-6-sulfonic acid).

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
