# Peer review of "Evaluating Antitumor and Antioxidant Activities of Yellow Monascus Pigments from Monascus ruber Fermentation"

_molecules, 2018, doi:10.3390/molecules23123242_

Round 1
Reviewer 1 Report
This is a clear and well-presented paper with some interesting results.
There needs to be a further round of English editing to improve the quality of the manuscript.
Some unrealistic claims are made about anti-breast cancer effects - these cell line studies are very far from demonstrating any therapeutic potential in vivo. More realistic writing on this issue is necessary.
Author Response
1: This is a clear and well-presented paper with some interesting results.
Response: Thank you very much for your kind evaluation.
2: There needs to be a further round of English editing to improve the quality of the
manuscript.
Response: Thanks. We have all the English language proofed in the revised
manuscript according to your suggestion.
3: Some unrealistic claims are made about anti-breast cancer effects - these cell line
studies are very far from demonstrating any therapeutic potential in vivo. More
realistic writing on this issue is necessary.
Response: Thanks a lot for your suggestion. This issue has been addressed in the
manuscript. Please find them in line 31-33 and 320-322 in the revised manuscript.

Reviewer 2 Report
In this study, the authors investigated the anti-tumor and anti-oxidant properties of water soluble natural yellow Monascus pigments (NYMPs). NYMPs were found to inhibit cell proliferation, cell migration and invasion of MCF-7 human breast cancer cells and reduce the expression of matrix metalloproteinases and vascular endothelial growth factor, key factors contributing to tumor metastasis and angiogenesis. NYMPs also exhibited more potent anti-oxidant property than the reduced yellow Monascus pigments (RYMPs) in the DPPH and ABTS assays. Overall most experiments were well-designed and conducted. Data presented in the manuscript were also properly organized and interpreted. This study highlights the anti-tumor and anti-oxidant potentials of NYMPs and is worth to be published.
Main concerns:
1. The cytotoxicity and cell proliferation studies.
Based on section 3.6, MTT assay was used for both studies. In cell proliferation assay, cells were only incubated for 4 h before treatment. Is 4 hours long enough for cell attachment?
Also, what medium was used for treatment in the cell proliferation assay? I assume it was the complete DMEM medium with FBS.
In this study, MTT might not be the best choice for cell proliferation. Assays such as BrdU would be a better choice which measures the DNA replication rates.
2. Constituents of NYMPs.
The authors referred to other publications on the NYMPs components. Any analysis (e.g. LC-MS) was performed in the current study? Would same Monascus ruber strain always produce similar fermentation products? RYMPs was compared with NYMPs for anti-oxidant potential. What are the major differences between the chemical profiles of the two products?
3. Western blot assay. Treatment details need to be provided, also the anti-body information can be added (manufacture, catalog no. etc.). What software was used for the quantification of western blot data?
4. Figures 1 and 6, error bar was missing for control groups.
5. Line 82, add “RYMPs” after “and”.
Author Response
1: The cytotoxicity and cell proliferation studies.
Based on section 3.6, MTT assay was used for both studies. In cell proliferation assay, cells were only incubated for 4 h before treatment. Is 4 hours long enough for cell attachment?
Also, what medium was used for treatment in the cell proliferation assay? I assume it was the complete DMEM medium with FBS.
In this study, MTT might not be the best choice for cell proliferation. Assays such as BrdU would be a better choice which measures the DNA replication rates.
Response:
Thanks a lot for your suggestion. In cell proliferation assay, cells were actually incubated for 24 h before treatment. We were very sorry for our carelessness and had the mistake corrected in the revised manuscript.
The H-DMEM medium containing FBS was used for treatment in the cell proliferation assay. The corrected part can be found in line 264 in the revised manuscript.
We agree well with that BrdU would be more suitable. We chose MTT assay by referring to a published paper [1]. Your suggestion is very conducive to our follow-up research and we will apply BrdU to our subsequent research.
2: Constituents of NYMPs.
The authors referred to other publications on the NYMPs components. Any analysis (e.g. LC-MS) was performed in the current study? Would same Monascus ruber strain always produce similar fermentation products?
RYMPs was compared with NYMPs for anti-oxidant potential. What are the major differences between the chemical profiles of the two products?
Response:
In our previous work, the molecular weights of Y1, Y2, Y3 and Y4 from Monascus ruber CGMCC10910 were 250, 254, 402 and 358, which were identified by LC–MS [2]. The same Monascus ruber strain will always produce similar fermentation products in the same fermentation conditions.
RYMPs are prepared from monascorubin, a kind of natured red Monascus pigments by reduced reaction. Monascorubin was converted to carboxylate by alkaline hydrolysis. Then, cyclic carbonyl group in monascorubin was reduced to hydroxyl group by specific oxygenated sulfides. NYMPs are the products directly extracted from Monascus fermentation broth. The major differences of the chemical profiles between two products are that the ingredients in RYMPs contain sulfonic acid group, which are added into NMPs (natured Monascus pigments) by reduced reaction. NYMPs don’t contain sulfonic acid group.
3: Western blot assay.
Treatment details need to be provided, also the anti-body information can be added (manufacture, catalog no. etc.). What software was used for the quantification of western blot data?
Response:
Thanks a lot for your suggestion. We had provided the treatment details in our manuscript in line 304-309 as below.
“visualized with use of a streptavidin-HRP ELC assay (Beyotime Institute of Biotechnology, Beijing, China). Western blotting involved anti-MMP 2, anti-MMP 9, anti-VEGFA and anti-GAPDH antibodies.” was replaced by “Membranes were blocked with 5% non-fat dry milk in TBS with 0.1% Tween-20 for 1 h and incubated with anti-MMP 2 (abcam, ab92536, 1:1000), anti-MMP 9 (abcam, ab76003, 1:1000), anti-VEGFA (Wanleibio, WL0009b, 1:1000) and anti-GAPDH (CST, #2118, 1:1000) antibodies at 4 °C overnight. The protein blots were then visualized with a use of the streptavidin-HRP ELC assay (Beyotime Institute of Biotechnology, Beijing, China). Image-Pro Plus 6.0 was used for the quantification of the data from western blot.”
4: Figures 1 and 6, error bar was missing for control groups.
Response:
Thank you for your suggestion. The error bars have been added in the figures. Please see the revised manuscript.
5: Line 82, add “RYMPs” after “and”.
Response:
Thanks for your careful reminder. We have corrected the mistake in the revised manuscript.
References
1. Huang, G.J.; Yang, C.M.; Chang, Y.S.; Amagaya, S.; Wang, H.C.; Hou, W.C.; Huang, S.S.; Hu, M.L. Hispolon Suppresses SK-Hep1 Human Hepatoma Cell Metastasis by Inhibiting Matrix Metalloproteinase-2/9 and Urokinase-Plasminogen Activator through the PI3K/Akt and ERK Signaling Pathways. J Agr Food Chem. 2010, 58, 9468-9475.
2. Huang, T.; Tan, H.L.; Chen, G.; Wang, L.; Wu, Z.Q. Rising temperature stimulates the biosynthesis of water-soluble fluorescent yellow pigments and gene expression in Monascus ruber CGMCC10910. AMB Express. 2017, 7, 134, doi:10.1186/s13568-017-0441-y.
